# The Relation Between Family Intimacy and Preschoolers’ Social–Emotional Competence: The Mediating Role of Psychological Resilience and the Moderating Role of Family–Preschool Interaction

**DOI:** 10.3390/bs15111564

**Published:** 2025-11-17

**Authors:** Haiping Wang, Xiaocen Liu

**Affiliations:** 1College of Education, Open University of China, Beijing 100039, China; wanghaip@ouchn.edu.cn; 2College of Preschool Education, Capital Normal University, Beijing 100048, China

**Keywords:** parent–child relationship, social–emotional development, emotional resilience, home–school partnership, preschool children, moderated mediation model

## Abstract

Social–emotional competence in early childhood is critical in shaping later academic achievement, interpersonal functioning, and long-term psychosocial development. While prior research has emphasized the importance of parenting practices, limited attention has been paid to how family dynamics and broader ecological interactions jointly contribute to this competence. Grounded in Social–Ecological Systems Theory, the present study investigated the association between family intimacy and preschoolers’ social–emotional competence, with psychological resilience as a mediator and family–preschool interaction as a moderator. A total of 2768 preschoolers aged 3 to 6 years from four provinces in China were assessed through parent-report questionnaires. Regression-based moderated mediation analyses revealed that family intimacy was positively associated with preschoolers’ social–emotional competence, directly and indirectly through psychological resilience. Moreover, family–preschool interaction moderated the pathway between family intimacy and psychological resilience, such that this association was stronger when family–preschool interactions were more frequent. These findings highlight the joint contribution of family relationships, individual strengths, and external developmental contexts to young children’s social–emotional development. Implications include supporting emotionally connected family environments, promoting resilience in early childhood, and strengthening home–preschool partnerships to foster holistic developmental outcomes.

## 1. Introduction

In recent years, research on social–emotional competence has gained increasing attention, particularly during the preschool years, characterized by rapid growth in emotional understanding, social behavior, and self-regulation ([41]; [35]). Social–emotional competence refers to a broad set of abilities that enable children to recognize and manage emotions, communicate effectively, cooperate with others, and pursue personal goals. It encompasses non-cognitive domains such as cognitive control, emotional expressivity, empathy, prosocial behavior, and emotional regulation ([29]; [31]). Developing these abilities relies on children’s growing capacity to express and regulate emotions appropriately and establish supportive relationships with adults and peers. Such competencies emerge and strengthen through both formal and informal learning experiences across early development ([12]; [56]). By four to five years of age, children begin to identify and interpret increasingly complex emotional states ([19]). Empirical evidence shows that higher levels of social–emotional competence in early childhood predict stronger peer relationships, greater academic readiness ([29]), and favorable long-term outcomes such as career success and higher income ([26]). Conversely, deficits in social–emotional competence—especially in emotion regulation—are associated with poorer interpersonal functioning, reduced social adjustment ([11]), and a higher likelihood of behavioral problems among children with developmental challenges such as autism spectrum disorder ([27]).

A growing body of research has examined various predictors of young children’s social–emotional development, including family-level characteristics such as parenting stress ([55]), parental health ([22]), parenting behaviors ([9]), and household income ([51]), as well as educational factors such as inclusive preschool practices ([57]). Among these family-related variables, one of the most salient is family intimacy, which refers to the emotional closeness and bonding among family members. It serves as an important indicator of the overall quality of family relationships and the affective climate within the home ([42]; [21]). However, empirical findings concerning the role of family intimacy in children’s social–emotional development remain mixed. Several studies have shown that close and supportive family relationships exert a strong positive influence on preschool children’s social–emotional growth ([58]; [59]). In contrast, a German study found that a warm emotional climate within the family did not significantly buffer children against emotional difficulties ([44]). Such inconsistencies suggest that the influence of family dynamics on children’s social–emotional competence may be shaped by additional contextual or individual factors ([66]). Consequently, further research is needed to clarify the mechanisms underlying how family intimacy contributes to preschoolers’ social–emotional competence, with particular emphasis on understanding how family processes interact with broader ecological systems to shape children’s developmental outcomes.

This study is grounded in Socio-Ecological Systems Theory ([10]), which provides the overarching framework for understanding how children’s development is shaped through dynamic interactions between individuals and their surrounding environments. Within this framework, micro-systems such as families and preschools directly influence children’s behavior and emotional growth. At the same time, the interconnections among these settings form meso-systems that further shape developmental outcomes. The theory underscores that human development is a co-constructed process, emerging from the continuous interplay between individuals and multiple ecological contexts. Building on this theoretical foundation, the present study also draws upon complementary perspectives, including family systems theory, attachment theory, developmental context theory, social capital theory, and social learning theory, to provide a more comprehensive understanding of children’s social–emotional development. Integrating these perspectives enables a multi-layered analysis that captures both internal and external mechanisms influencing development. Guided by this approach, the study examines how family intimacy relates to preschoolers’ social–emotional competence, focusing specifically on the mediating role of psychological resilience and the moderating role of family–preschool interaction.

### 1.1. The Family Intimacy and Preschoolers’ Social–Emotional Competence

Family intimacy refers to the emotional closeness and bonding experienced among family members ([42]). Family systems theory indicates that intimacy influences the externalization of family systems effects ([20]). Within intimate family environments, children can seek emotional support and guidance from caregivers when distressed, and they also tend to internalize emotional patterns modeled by their parents ([45]). Empirical studies have reported that positive parent–child relationships may help buffer children against negative influences from parents, family, or other sources of adversity, keeping stress responses within a “tolerable” range. ([61]). In contrast, disruptions in parent–child relationships have been linked to compromised social–emotional functioning in both infancy and early childhood ([49]; [64]).

Attachment Theory provides a conceptual basis for understanding family intimacy. Secure attachment—often viewed as a core expression of family intimacy—has been found to predict more effective emotional regulation in young children ([38]). The parent–child relationship forms the foundation for children’s Internal Working Models, which guide their expectations and behaviors in social relationships ([1]). Adverse experiences, such as early parent–child separation, may interfere with forming secure attachments and lead to difficulties in emotional bonding, potentially contributing to maladaptive personality traits and social difficulties in later development ([60]).

Taken together, these findings suggest that emotionally cohesive and responsive family relationships may play a crucial role in the development of children’s emotional understanding and interpersonal competence. Accordingly, it is expected that higher levels of family intimacy will be associated with greater social–emotional competence among preschool children.

### 1.2. The Mediating Role of Psychological Resilience

Psychological resilience denotes an individual’s capacity to adapt flexibly to adversity and to recover from stressful or negative experiences ([46]). Beyond its role as a personal capacity, psychological resilience has also been conceptualized as a developmental process, reflecting “normal development under difficult conditions” ([43]). In early childhood, the family environment is critical in shaping psychological resilience, functioning as both a proximal support system and a buffer against external stressors. According to the Conservation of Resources Theory, children’s behavioral and emotional development is influenced by the interaction between external resources (e.g., family relationships) and internal traits (e.g., resilience) ([25]). As a key relational resource, the parent–child relationship has been identified as an important contextual factor associated with the development of resilience in preschool-aged children.

Attachment-based perspectives also emphasize the role of sustained caregiver relationships in constructing internal coping frameworks. Through repeated interactions with caregivers, children develop Internal Working Models that influence their beliefs about self-worth, emotion regulation strategies, and responses to challenges ([37]). Empirical evidence has shown that parent–child closeness is positively associated with preschoolers’ psychological resilience ([39]; [18]), and that emotional warmth and responsive parenting may promote adaptive coping in both normative and high-risk developmental contexts ([3]).

Resilient children are often better equipped to navigate social demands, regulate emotions, and sustain positive peer relationships ([69]; [5]). These attributes closely align with the foundational components of social–emotional competence. Empirical studies have demonstrated that psychological resilience is negatively associated with emotional and behavioral problems such as impulsivity, hyperactivity, and anxiety ([7]; [5]). Moreover, a positive family environment—characterized by emotional warmth and secure attachments—can promote the natural development of resilience in preschoolers by fulfilling their basic psychological needs, such as a sense of safety and belonging. Such a supportive environment, in turn, serves a protective role by buffering the adverse effects of external stressors and reducing the likelihood of emotional and behavioral difficulties ([63]).

Building on these insights, resilience can be viewed as a core mechanism through which emotionally intimate family relationships contribute to adaptive socio-emotional functioning. Therefore, it is anticipated that psychological resilience will serve as a key mechanism linking family intimacy to preschoolers’ social–emotional competence.

### 1.3. The Moderating Role of Family–Preschool Interaction

Family–preschool interaction refers to the coordinated efforts between families and preschools to promote children’s development. It encompasses parental involvement in preschool education and preschool guidance for family education, forming the foundation of effective home–school collaboration ([24]). According to Developmental Context Theory, children’s development arises from the dynamic interplay among multiple microsystems—such as the family and preschool—rather than being shaped by any single context in isolation ([33]). Both family and preschool environments thus play essential and complementary roles in shaping early social–emotional functioning. Within this framework, the association between family intimacy and children’s internal strengths—such as psychological resilience—may vary depending on the quality and frequency of family–preschool interaction, highlighting the importance of cross-contextual collaboration in early development.

Empirical evidence indicates that stronger family–preschool collaboration is linked to improved parenting practices, enhanced parent–child interactions, and a more supportive foundation for children’s emotional and behavioral development ([47]; [2]). Children who experience frequent and high-quality communication between families and preschools tend to exhibit greater emotional competence and more advanced social skills during early childhood ([62]). Furthermore, research has shown that positive teacher–child and peer relationships can buffer the negative impact of adverse family environments on the development of children’s psychological resilience ([50]; [15]). Similarly, consistent caregiving practices across family and preschool contexts have been associated with stronger coping abilities among young children ([67]).

Collectively, these findings underscore the importance of cross-contextual coordination between home and school in shaping early developmental outcomes. Thus, it is plausible that the strength of the association between family intimacy and psychological resilience varies according to the level of family–preschool interaction.

### 1.4. The Present Study

Although numerous studies have confirmed the positive associations between family environment, family intimacy, and children’s social–emotional development, research specifically targeting the preschool period remains limited in scope and depth. Most prior studies have focused on single contexts or small regional samples, leaving the broader ecological interplay among family, individual, and educational systems insufficiently examined. Moreover, while previous research has investigated the independent influences of family and preschool contexts—and some studies have highlighted the benefits of home–preschool collaboration for children with special educational needs—few have integrated micro-contextual factors (e.g., family intimacy), internal mechanisms (e.g., psychological resilience), and cross-contextual processes (e.g., family–preschool interaction) within a unified analytical framework.

To fill this gap, the present study utilizes a large, multi-province sample of typically developing preschoolers in China to examine how family intimacy relates to children’s social–emotional competence through psychological resilience and whether this indirect pathway varies according to the level of family–preschool interaction. This integrative, moderated mediation framework advances previous work by capturing both within-family and cross-contextual processes, thereby providing a more comprehensive understanding of preschoolers’ social–emotional development within the Chinese socio-educational context.

Building on these aims, the present study investigates how family intimacy relates to preschoolers’ social–emotional competence, with a particular focus on the mediating role of psychological resilience and the moderating role of family–preschool interaction.

To further clarify the scope of inquiry, this study seeks to answer the following research questions: How is family intimacy associated with preschoolers’ social–emotional competence? Does psychological resilience mediate the relationship between family intimacy and preschoolers’ social–emotional competence? Moreover, does the strength of this indirect association vary according to the level of family–preschool interaction?

Based on these research questions and the theoretical framework discussed above, the following hypotheses are proposed:

**Hypothesis** **1.***There is a positive correlation between family intimacy and preschoolers’ social–emotional competence. It is postulated that closer emotional bonds within the family may correspond to higher levels of social–emotional functioning in early childhood*.

**Hypothesis** **2.***Psychological resilience mediates the association between family intimacy and preschoolers’ social–emotional competence. It is posited that higher family intimacy may correspond to enhanced resilience, which in turn may relate to more competent emotional and social functioning*.

**Hypothesis** **3.***Family–preschool interaction is hypothesized to moderate the association between family intimacy and preschoolers’ psychological resilience. It is further suggested that higher levels of interaction between families and preschools may be linked to a stronger relation between family intimacy and resilience*.

The conceptual framework is presented in Figure 1.

## 2. Materials and Methods

### 2.1. Participants

Convenience sampling was employed to recruit preschool-aged children from four provinces in China. Because preschoolers are unable to complete self-report questionnaires accurately, their parents (primary caregivers)—who serve as the primary caregivers and possess comprehensive knowledge of their children’s daily behaviors and emotional states—were invited to complete the survey on behalf of their children. This parent-report (observer-rated) approach ensured both the reliability and ecological validity of the collected data.

A total of 2800 questionnaires were distributed, yielding 2768 valid responses and an effective response rate of 98.9%. Among the participating children, 1450 (52.4%) were boys and 1318 (47.6%) were girls. The children’s ages ranged from 3 to 6 years (*M* = 4.72, *SD* = 0.87). Regarding family structure, 778 children (28.1%) were only children, whereas 1990 (71.9%) had siblings. In terms of residence, 2096 respondents (75.7%) lived in urban areas, while 672 (24.3%) resided in rural regions. The study protocol and procedures received approval from the Institutional Review Board at the authors’ home institutions, and informed consent was obtained from all participating caregivers.

All participants were typically developing preschoolers. Children with diagnosed developmental disorders (e.g., autism spectrum disorder, attention-deficit/hyperactivity disorder), psychiatric conditions, or major family crises (e.g., parental bereavement) were excluded based on parent reports and preschool records. Inclusion criteria required that participating children be enrolled full-time in preschool and living with at least one parent or primary caregiver. Exclusion criteria were applied during data collection to ensure the validity of responses.

### 2.2. Measures

#### 2.2.1. Family Intimacy

The intimacy dimension of the Child–Parent Relationship Scale Short-Form (CPRS-SF; [16]; [14]) was used to operationalize family intimacy. This subscale contains seven items (e.g., “I have a caring, warm relationship with my child”; “If upset, my child comes to me for comfort”) rated on a five-point Likert scale (1 = not at all; 5 = completely). An average score was calculated, with higher values indicating greater parent–child emotional closeness. In the current sample, Cronbach’s α = 0.70. The CPRS-SF closeness subscale has demonstrated robust internal consistency and convergent validity across preschool-aged populations ([16]; [48]).

#### 2.2.2. Social–Emotional Competence

Preschoolers’ social–emotional competence was assessed using the Chinese Inventory of Children’s Social–emotional Competence (CICSEC; [36]), which includes 30 items covering emotion regulation, empathy, cognitive control, and prosocial behavior. Items (e.g., “Easily forgets longer instructions”; “After completing a task, forgets the task that follows”) were rated on a five-point Likert scale (1 = strongly disagree; 5 = strongly agree). After reverse-scoring relevant items, mean scores were computed so that higher values reflect higher levels of competence. In this study, Cronbach’s α = 0.90. The CICSEC has shown excellent internal consistency and criterion validity in Chinese preschool samples, with positive correlations to school readiness and negative correlations with behavior problems ([35]).

#### 2.2.3. Psychological Resilience

Psychological resilience was measured using the Chinese version of the Devereux Early Childhood Assessment for Preschoolers, Second Edition (DECA-P2; [32]; translated by [28]). The scale consists of 27 items (e.g., “My child behaves in a way that makes adults laugh or show interest in him/her”; “My child listens to or respects others”) rated on a five-point scale from 0 (“not at all”) to 4 (“completely”). After reverse scoring as needed, mean scores were calculated, with higher scores reflecting greater resilience. Cronbach’s α in this sample was 0.95.

#### 2.2.4. Family–Preschool Interaction

Family–preschool interaction was captured via a self-report item adapted from prior field surveys: “How often have you interacted with the preschool in the past year (including but not limited to communicating with teachers, participating in preschool activities, volunteering as a parent, etc.)?” Responses were rated on a five-point Likert scale (1 = very low, 5 = very high), with higher scores representing greater family–preschool engagement. Although measured via a single item, this indicator reflects observable behavioral engagement and aligns with prior research using frequency-based interaction measures ([60]).

### 2.3. Data Processing

The data analysis proceeded in three stages. First, after data entry into SPSS version 23.0 (IBM Corp., Armonk, NY, USA), an unrotated exploratory factor analysis was conducted to perform Harman’s single-factor test, which assessed common method bias by examining whether a single factor accounted for the majority of variance among all measurement items. Second, Pearson correlation analyses and comparative analyses were conducted to examine the associations and group differences among the primary study variables, providing preliminary insights into their interrelationships and demographic variations. Third, conditional process analysis was conducted using the PROCESS macro (version 4.1) for SPSS, developed by Andrew F. Hayes. To test the hypothesized mediation model, Model 4 was employed, examining the mediating role of psychological resilience in the association between family intimacy and preschoolers’ social–emotional competence. To further test the moderating role of family–preschool interaction, Model 7 was applied, in which the moderator operated on path a (the effect of family intimacy on psychological resilience). A moderated mediation model was then estimated to assess whether the indirect effect of family intimacy on social–emotional competence via psychological resilience varied as a function of family–preschool interaction. All analyses controlled for children’s age, gender, only-child status, and residential location. The significance of indirect and conditional effects was tested using bias-corrected bootstrap confidence intervals based on 5000 resamples, following recommended practices for conditional process modeling ([23]). This analytic approach follows contemporary best practices for testing complex mediation and moderation models within a regression-based framework.

### 2.4. Common Method Biases

To mitigate potential common method bias, procedural controls were implemented, including anonymous participation and reverse-coded items. To further enhance methodological rigor, Harman’s single-factor test was conducted. The results revealed that 12 factors had eigenvalues greater than 1, and the first factor accounted for only 24.14% of the total variance, below the commonly accepted threshold of 40%. These findings suggest that common method bias was not a serious concern in this study.

## 3. Results

### 3.1. Descriptive Statistics and Correlation Analysis

Table 1 presents the results of the correlation analyses. Family intimacy was positively associated with psychological resilience, family–preschool interaction, and social–emotional competence. Psychological resilience was also positively associated with family–preschool interaction and social–emotional competence. Furthermore, family–preschool interaction positively correlated with social–emotional competence. Given that age was significantly correlated with the core variables, it was included as a control variable in the subsequent analyses.

### 3.2. Comparative Analysis

After presenting the overall descriptive statistics and correlations among the key study variables, group comparisons were conducted to examine whether family intimacy, psychological resilience, family–preschool interaction, and social–emotional competence differed by gender, only-child status, and family residence. The results of these comparative analyses are summarized in Table 2.

Significant gender differences were observed in family intimacy, psychological resilience, and social–emotional competence, with girls scoring significantly higher than boys across these three domains. However, no gender difference emerged for family–preschool interaction.

Regarding family structure, significant differences were identified in family–preschool interaction, with non-only children reporting higher levels of interaction than only children. No significant differences were found in family intimacy, psychological resilience, or social–emotional competence between these two groups.

Differences were also evident across residential backgrounds. Across all variables—family intimacy, psychological resilience, family–preschool interaction, and social–emotional competence—children from urban families scored higher than those from rural families. These findings suggest that urban environments may provide more supportive emotional and educational contexts, facilitating family relationships, resilience development, and children’s social–emotional functioning.

Accordingly, gender, only-child status, and residential background were included as control variables in subsequent analyses to account for their potential influence on the primary relationships among family intimacy, psychological resilience, and social–emotional competence.

### 3.3. Moderated Mediation Effects Analysis

After standardizing all variables, a mediation model was first tested using PROCESS Model 4 in SPSS (Version 23.0; [23]), with children’s age, gender, only-child status, and residential location included as covariates. The results indicated that family intimacy positively predicted psychological resilience (β = 0.61, *SE* = 0.15, *p* < 0.001) and social–emotional competence (β = 0.17, *SE* = 0.02, *p* < 0.001). Additionally, psychological resilience positively predicted social–emotional competence (β = 0.27, *SE* = 0.02, *p* < 0.001). The indirect effect of family intimacy on social–emotional competence through psychological resilience was significant, with an effect size of 0.16 and a bias-corrected 95% bootstrap confidence interval [0.127, 0.196] (based on 5000 resamples), which did not include zero. These results suggest that the emotional closeness between family members may help preschoolers develop greater psychological resilience, which is associated with better social–emotional functioning performance.

To examine the moderating role of family–preschool interaction, a moderated mediation model was estimated using PROCESS Model 7 ([23]), in which the moderator operated on path a (i.e., the effect of family intimacy on psychological resilience). Children’s age, gender, only-child status, and residential location were again included as covariates. As shown in Table 3, the interaction between family intimacy and family–preschool interaction positively predicted psychological resilience (β = 0.06, *SE* = 0.01, *p* < 0.001), indicating a significant moderating effect. The index of moderated mediation, based on 5000 bias-corrected bootstrap samples, was 0.017 (95% CI [0.002, 0.032]), excluding zero. These findings indicate that the indirect effect of family intimacy on social–emotional competence through psychological resilience varies significantly across different levels of family–preschool interaction.

To further explore the nature of this moderated mediation, we conducted a simple-slope analysis by dividing the standardized family–preschool interaction scores into two groups: high (+1 SD) and low (−1 SD). Separate analyses were conducted for each group, and the results are presented in Table 4. The indirect effect of family intimacy on social–emotional competence, mediated by psychological resilience, was significant at both interaction levels. Specifically, at low family–preschool interaction, the indirect effect was 0.14 (95% CI [0.108, 0.174]); at high interaction, the effect was more substantial (0.17; 95% CI [0.135, 0.214]), both confidence intervals excluding zero. These findings indicate that the mediating role of psychological resilience remains robust across interaction levels, with a slightly stronger indirect association under high family–preschool interaction.

Further simple-slope analyses revealed that the strength of the association between family intimacy and psychological resilience varied as a function of family–preschool interaction (see Figure 2). At low family–preschool interaction, family intimacy significantly predicted psychological resilience (β = 0.53, *t* = 26.53, *p* < 0.001), whereas at high interaction the predictive effect was stronger (β = 0.65, *t* = 31.53, *p* < 0.001). Notably, this pattern suggests that close and frequent collaboration between families and preschools may amplify the positive developmental influence of family intimacy on children’s psychological resilience.

## 4. Discussion

This study advances existing research by addressing a key conceptual gap in the literature. Whereas prior studies have typically examined the effects of family and preschool environments in isolation—or focused primarily on children with special educational needs—this study integrates family intimacy (as a micro-contextual factor), psychological resilience (as an internal mechanism), and family–preschool interaction (as a cross-contextual moderator) within a unified moderated mediation model.

By situating these constructs within the Chinese cultural and educational context and using a large, multi-province sample of typically developing preschoolers, the study not only extends previous findings but also demonstrates how multi-systemic processes jointly shape preschoolers’ social–emotional competence. This integrative framework presents a novel, multi-level ecological perspective that enhances our understanding of how familial and educational systems interact to support children’s socio-emotional development.

### 4.1. Associations Between Family Intimacy and Preschoolers’ Social–Emotional Competence

Empirical evidence from this study indicated that family intimacy was positively associated with preschoolers’ social–emotional competence ([38]; [59]). This result not only confirms earlier findings but also extends their implications by illustrating how the quality of family emotional bonds contributes to children’s broader socio-emotional adaptation in early childhood.

This finding aligns with both Family Systems Theory and Social Learning Theory. According to Family Systems Theory, emotional exchanges within the family are interdependent and may be transmitted across generations, shaping the emotional climate in which children develop ([40]). Warm, emotionally cohesive family environments contribute to children’s emotional health and overall well-being, whereas deficiencies in parental affection—whether maternal or paternal—pose significant risks to the development of children’s social–emotional competence ([17]). From this perspective, family intimacy serves as an emotional foundation that enables children to internalize secure attachment patterns and to develop empathy, cooperation, and effective emotion regulation.

From the perspective of Social Learning Theory, individuals acquire behavioral and emotional competencies through direct experience and observational learning ([6]). Within intimate and harmonious families, preschoolers are continually exposed to consistent models of adaptive social behavior, which they observe, imitate, and gradually internalize. Such repeated interactions serve as natural learning opportunities that strengthen their ability to form and maintain positive relationships. Moreover, this process enhances social cognition—children learn to perceive others’ emotions accurately, interpret social cues, and respond appropriately to different interpersonal situations. Thus, family intimacy not only facilitates observational learning but also provides an emotionally safe environment in which children can practice and refine prosocial behaviors.

Empirical evidence further supports these associations. Close parent–child relationships are linked to the maintenance of positive affective states and the acquisition of fundamental social skills, which together reduce the likelihood of behavioral difficulties in early childhood ([8]). Conversely, research from Canada has demonstrated that repeated interruptions in parent–child interactions—such as those caused by parental smartphone use—can impair the development of preschoolers’ social–emotional competence by disrupting emotional attunement and communication ([12]). Taken together, these findings underscore that family intimacy is not merely a relational condition but a developmental context through which emotional understanding, regulation, and social participation are continuously cultivated.

### 4.2. The Mediating Role of Psychological Resilience

This study identified a significant positive association between family intimacy and psychological resilience, consistent with previous research ([3]; [18]; [39]). In turn, psychological resilience significantly predicted preschoolers’ social–emotional competence, echoing prior findings ([5]; [69]). Together, these results confirmed the mediating role of psychological resilience in the association between family intimacy and social–emotional competence. This pattern of findings suggests that emotionally cohesive family environments not only offer immediate socio-emotional support but also cultivate children’s capacity to recover from setbacks, adapt to challenges, and regulate emotions effectively over time.

From a theoretical standpoint, this relationship can be interpreted through the Dynamic Model of Psychological Resilience ([34]) and the Conservation of Resources Theory ([53]). Within these frameworks, psychological resilience is conceptualized as a dynamic process involving the continuous interaction between internal strengths and external protective factors. It functions as a regulatory mechanism that mitigates the impact of adverse life events. Accordingly, it is essential to consider the interplay among multiple risk and protective processes that exert varying degrees of influence on developmental outcomes ([43]). In this view, resilience emerges not as a fixed personality trait but as an adaptive capacity shaped through reciprocal exchanges between children and their social environments. From these theoretical perspectives, children draw upon external resources—such as family intimacy—to satisfy their fundamental psychological needs for love, belonging, and security. When these needs are consistently met, children are more likely to develop internal psychological resources, including resilience, that foster adaptive coping and overall well-being.

On one hand, the contribution of family intimacy to preschoolers’ psychological resilience is mainly attributable to experiences of parental trust. Within trusting and emotionally responsive parent–child interactions, children feel understood, valued, and respected—an experience that constitutes a powerful psychological resource. Such trust enhances children’s confidence in their competence and supports the formation of resilient coping patterns when facing challenges. In addition, open and supportive communication within the family further reinforces children’s belief systems and positive coping orientations. Through consistent, high-quality exchanges with parents, children learn to express their needs effectively, interpret emotions accurately, and adopt adaptive strategies to manage everyday stressors ([37]). Over time, these experiences consolidate a stable sense of emotional security and self-efficacy, both of which are fundamental building blocks of resilience. Moreover, close and affectionate parent–child relationships can serve as a protective buffer, mitigating the negative impact of parental psychological distress—such as anxiety or depression—on children’s development of resilience ([54]).

On the other hand, psychological resilience promotes the development of social–emotional competence. When encountering environmental challenges or stress, preschoolers with higher levels of resilience are more likely to adopt proactive and adaptive coping strategies. Such responses facilitate better social functioning and reduce the likelihood of internalizing behavioral problems ([65]; [4]). Moreover, resilient children tend to maintain emotional balance in social interactions, display empathy toward peers, and resolve conflicts constructively—skills that are central to social–emotional competence. This interplay suggests that resilience not only mediates the link between family intimacy and social–emotional competence but also reinforces children’s capacity for positive adaptation within emotionally supportive families.

### 4.3. The Moderating Role of Family–Preschool Interaction

Beyond the mediating pathway identified above, the present study also revealed that family–preschool interaction functioned as a significant contextual moderator, amplifying the positive influence of family intimacy on children’s psychological resilience. This cross-contextual process highlights that developmental outcomes are shaped not by isolated systems but through the synergistic functioning of family and preschool environments. The finding aligns with previous research emphasizing the joint operation of protective factors across multiple ecological levels ([43]; [62]; [67]).

Consistent with the Socio-Ecological Systems Theory and Developmental Contextualism, these results demonstrate that the influence of one relational context (e.g., the family) depends on the quality of another (e.g., the preschool), reflecting the dynamic interplay among microsystems that collectively shape children’s adaptation. Identifying this mechanism extends existing scholarship by shifting the research focus beyond the traditional emphasis on family environments alone or family–school collaboration among children with special educational needs ([66]; [2]).

Specifically, the family–preschool interaction moderated the effect of family intimacy on preschoolers’ social–emotional competence through psychological resilience. Among families reporting higher levels of home–school collaboration, family intimacy demonstrated a stronger predictive effect on children’s resilience. Consequently, the indirect effect of family intimacy on social–emotional competence through psychological resilience was more pronounced. In contrast, this mediating effect was weaker for families with lower family–preschool interaction. This finding supports the Enhancement Model of Interaction, which posits that one relational resource can amplify the positive effect of another in reducing children’s problematic behaviors and promoting development ([52]). Thus, when parents and educators maintain consistent and collaborative communication—through home visits, class activities, or feedback exchanges—they co-construct a cohesive support system that strengthens children’s emotional regulation and adaptive functioning.

This mechanism can be further understood through the lens of Social Capital Theory, which posits that cooperation and feedback within social networks shape individual behavior ([13]). Families and preschools represent two primary socialization systems where children accumulate social capital. High-quality collaboration between families and preschools increases parents’ social capital by fostering intergenerational closure—a reciprocal network characterized by shared norms, mutual trust, and open communication. Such networks facilitate the exchange of information about children’s development, strengthen coordination between home and school, and ultimately create a more coherent and supportive environment for children’s emotional and social growth ([30]; [68]).

In addition, prior studies have shown that effective family–preschool communication enables parents to adjust their parenting strategies according to children’s evolving socio-emotional needs, while allowing teachers to better understand each child’s unique temperament and family context. This bidirectional collaboration fosters continuity between home and preschool experiences, leading to consistent emotional expectations, smoother transitions, and ultimately stronger social–emotional competence ([47]).

### 4.4. Limitations and Future Directions

Despite the contributions of this study, several limitations should be acknowledged. First, the cross-sectional design limits the ability to draw causal inferences about the associations among family intimacy, psychological resilience, and preschoolers’ social–emotional competence. While the moderated mediation model provides theoretical insights, it is impossible to fully clarify the observed relations’ directionality. Future studies should adopt longitudinal or experimental designs to validate the temporal sequence and causal mechanisms implied by the current findings.

Second, the study relied exclusively on quantitative data, which limits the depth of understanding of how these factors interact in daily family and preschool contexts. Incorporating qualitative or mixed-method approaches—such as parent and teacher interviews, naturalistic classroom observations, or case studies—would allow researchers to capture the nuanced interpersonal and contextual processes underlying these associations and provide a more comprehensive picture of children’s developmental dynamics.

Third, the family–preschool interaction measurement relied on a single-item self-report assessing family–preschool interaction. Although this measure captures the overall level of collaboration between families and preschools, it may not fully reflect the multidimensional nature of such interactions, including the quality, content, and reciprocity of communication. Future research should incorporate multidimensional scales or mixed-method approaches (e.g., parent interviews, teacher reports, and observational data) to provide a more comprehensive family–preschool collaboration assessment.

Fourth, the study adopted a one-way evaluation design relying solely on parent-reported questionnaires, which may be subject to social desirability and reporting biases. Incorporating multiple informants, such as teachers or independent observers, and combining subjective reports with objective behavioral assessments would improve the validity and robustness of future findings.

Finally, this study did not include other potentially important contextual variables, such as parental mental health, parenting styles, or peer relationships, which may also influence preschoolers’ social–emotional competence. Future research could adopt more integrative models that account for multiple family, school, and community-level factors to better capture the complex interplay of influences on early childhood development.

### 4.5. Implications

This study integrates Social–Ecological Systems Theory and Developmental Context Theory to examine how family intimacy contributes to preschoolers’ social–emotional competence. By highlighting the mediating role of psychological resilience (an internal developmental resource) and the moderating role of family–preschool interaction (an external contextual factor), the findings provide empirical support for the cultural applicability of these theoretical frameworks within the Chinese early childhood context.

The results underscore the importance of fostering strong emotional bonds within the family and active home–school partnerships to support children’s adaptive development. Specifically, parents should take primary responsibility for cultivating a warm and trusting parent–child relationship and promoting psychological resilience by encouraging autonomy, emotional regulation, and problem-solving after children encounter setbacks ([25]). Educators, in turn, should maintain regular and high-quality communication with parents—especially those from separated or less-involved families—to provide timely feedback about children’s emotional and social needs. This collaborative approach facilitates alignment between parenting practices and educational objectives, enhances children’s internal coping capacities, and reinforces consistent support across home and school settings ([47]).

These findings suggest that cultivating preschoolers’ social–emotional competence requires a holistic, systemic, and synergistic effort between families and early childhood education institutions. Future policies and interventions should promote integrated family–school engagement models that recognize the dynamic interplay between internal and external developmental resources.

## 5. Conclusions

This study examined the mechanisms linking family intimacy to preschoolers’ social–emotional competence, with psychological resilience as a mediator and family–preschool interaction as a moderator. The results demonstrated that family intimacy positively predicted preschoolers’ social–emotional competence, and this association was partially mediated by psychological resilience. Moreover, family–preschool interaction significantly moderated the first stage of the mediation pathway—specifically, the link between family intimacy and psychological resilience. The indirect effect of family intimacy on social–emotional competence via psychological resilience was stronger when family–preschool interaction occurred more frequently. These findings highlight a moderated mediation mechanism through which internal and external resources jointly shape early social–emotional development.

## Figures and Tables

**Figure 1 behavsci-15-01564-f001:**
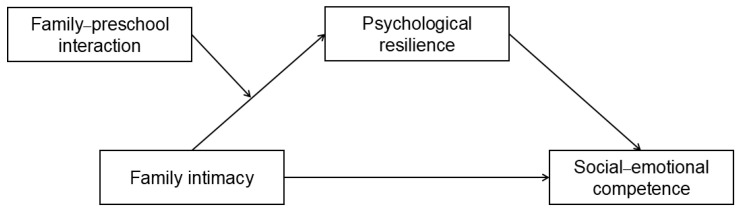
Hypothesized moderated mediation model.

**Figure 2 behavsci-15-01564-f002:**
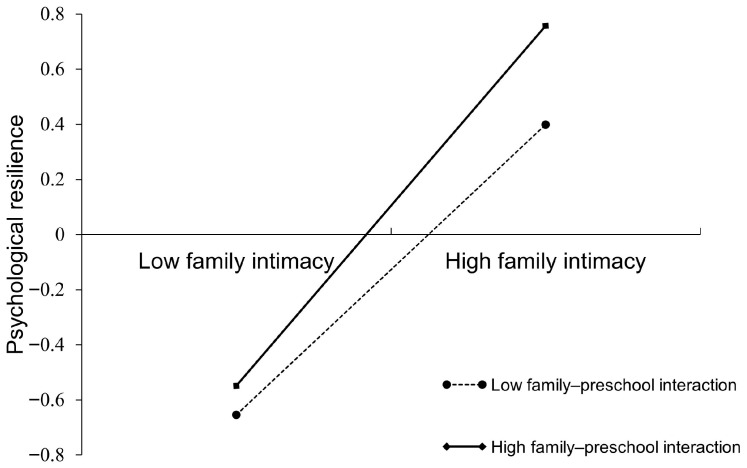
Moderating effect of family–preschool interaction on the link between family intimacy and psychological resilience.

**Table 1 behavsci-15-01564-t001:** Descriptive and correlation analysis.

Variables	*M*	*SD*	1	2	3	4	5
1. Age	4.72	0.87	1				
2. Family intimacy	3.51	0.31	0.003	1			
3. Psychological resilience	3.87	0.55	−0.001	0.612 ***	1		
4. Family–preschool interaction	3.85	1.07	0.000 †	0.186 ***	0.221 ***	1	
5. Social–emotional competence	3.56	0.50	−0.063 **	0.337 ***	0.379 ***	0.130 ***	1

† 0.000 indicates no detectable correlation. ** *p* < 0.01, *** *p* < 0.001.

**Table 2 behavsci-15-01564-t002:** Differential analysis of demographic variables.

Variables	Family Intimacy	Psychological Resilience	Family–Preschool Interaction	Social–Emotional Competence
Gender				
Boy (n = 1450)	3.50 ± 0.31	3.84 ± 0.54	3.84 ± 1.07	3.52 ± 0.48
Girl (n = 1318)	3.53 ± 0.31	3.90 ± 0.55	3.87 ± 1.08	3.61 ± 0.52
*t*	−2.099 *	−2.926 **	−0.773	−5.049 ***
Only-child status				
Only child (n = 778)	3.51 ± 0.31	3.86 ± 0.52	3.78 ± 1.08	3.53 ± 0.48
Non-only child (n = 1990)	3.51 ± 0.31	3.87 ± 0.55	3.88 ± 1.07	3.57 ± 0.51
*t*	−0.021	−0.466	−2.389 *	−1.923
Residential location				
Rural (n = 672)	3.47 ± 0.33	3.77 ± 0.65	3.77 ± 1.15	3.46 ± 0.50
Urban (n = 2096)	3.53 ± 0.30	3.90 ± 0.51	3.88 ± 1.05	3.59 ± 0.50
*t*	−4.139 ***	−4.857 ***	−2.098 *	−5.796 ***

* *p* < 0.05, ** *p* < 0.01, *** *p* < 0.001.

**Table 3 behavsci-15-01564-t003:** Test of the moderating role of family–preschool interaction.

Variables	β	*SE*	*t*	95%CI	
				Lower	Upper
Gender	0.06	0.03	1.93	−0.001	0.115
Age	−0.00	0.02	−0.07	−0.035	0.033
Only child status	0.02	0.03	0.57	−0.047	0.085
Residential location	0.13	0.03	3.60 ***	0.057	0.194
Family intimacy	0.59	0.02	38.99 ***	0.560	0.619
Family–preschool interaction	0.12	0.02	7.64 ***	0.086	0.145
Family intimacy × Family–preschool interaction	0.06	0.01	4.70 ***	0.037	0.090
*R* ^2^			0.40 ***		
*F*			257.91 ***		

*** *p* < 0.001.

**Table 4 behavsci-15-01564-t004:** Conditional indirect effects via psychological resilience at varying family–preschool interaction levels.

Mediating Variable	Moderator Level	Indirect Effect	Boot SE	95% CI Lower	95% CI Upper
Psychological resilience	Low family–preschool interaction	0.14	0.02	0.108	0.174
	High family–preschool interaction	0.17	0.02	0.135	0.214

## Data Availability

The data supporting the findings of this study are available from the authors upon reasonable request.

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
