# Peer review of "The Relation Between Family Intimacy and Preschoolers’ Social–Emotional Competence: The Mediating Role of Psychological Resilience and the Moderating Role of Family–Preschool Interaction"

_behavsci, 2025, doi:10.3390/bs15111564_

Round 1

Reviewer 1 Report

Comments and Suggestions for Authors

Thank you for the opportunity to review the article” The Relation between Family Intimacy and Preschoolers’ Social–Emotional Competence: The Mediating Role of Psychological Resilience and the Moderating Role of Family–Preschool Interaction.”

Here are my comments for the authors to improve their manuscript

  • The Abstract should follow the journal's guidelines (there are no subsections for the Abstract, and Methods are missing completely from the Abstract)
  • The title has some keywords that must be found and explained in the Introduction section. (Family Intimacy, Social–Emotional Competence, Psychological Resilience, and the Family–Preschool Interaction)
  • The hypothesis should be formulated at the end of the Introduction section, together with the aim of the study first. 
  • Citations in the manuscript should conform to the journal's rules
  • References should be reported conforming to the journal's rules.
  • Descriptive statistics, correlation, comparative, and at the end, mediation/moderation.
  • IN the Discussion section, the results must be presented in congruence or in opposition with results from the literature. Sustaining the results with only one scientific article, the majority of them from the same country, is not a manner in which the authors could sustain that their results are generalizable. 

Reviewer 2 Report

Comments and Suggestions for Authors

see file attached

Reviewer 3 Report

Comments and Suggestions for Authors
  • The literature review is generally clearly written, informed by relevant theories and studies which support the hypotheses well. Eventhough the hypotheses are typically presented at the end of the entire section, I am keen to see this at the end of each of the respective sub-section which you have done so. As you had reported previous studies which found your hypotheses to be true, what is the aim or intent for researching for each factor? You would need to clearly point out the gaps in each of these factors or in the context you are investigating.
  • Another thing to add at the end of the literature review will be the research question(s) under section 1.4 on page 4 which will provide a clear recap of the inquiry and aims of the study.
  • The methods were well-selected and complemented the theories and hypotheses raised in literature review.
  • Section 3.1 on page 6 should be shifted to section 2.
  • The findings and discussion were clearly presented. However, since all three findings align with previous studies, the results were not surprising. What is the specific gap that this study is filling up? Is this the first study in this particular context? How is the context in your study differing from other studies which had been reported from the Chinese demographics? How was the selection of measures differ from existing studies? What are other areas which could be different from existing studies which make up the rationale for your investigation?
  • In addition, if there were some qualitative data from your surveys, it would make the analyses more interesting if the specifics for each factor were reported. For example, what contributes to closeness (intimacy) between parent and child? If there is no qualitative data, there needs to be a robust rationale for the intent and gap of this study.

Round 2

Reviewer 1 Report

Comments and Suggestions for Authors

The version of the manuscript has really improved. But some small points must be clarified:

  • The instruments were addressed to parents or teachers, and this information must be clearly specified in the Materials and Methods section.
  • The one-way evaluation must be included among the study's limitations.
  • Inclusion/exclusion criteria are extremely important - are psychological, psychiatric, social, family-related issues taken into consideration?
  • Are all keywords from the title developed into the Introduction, Results, and Discussion sections? Please verify.

Round 3

Reviewer 1 Report

Comments and Suggestions for Authors

Thank you for the new version of the manuscript entitled ”The Relation between Family Intimacy and Preschoolers’ Social–Emotional Competence: The Mediating Role of Psychological Resilience and the Moderating Role of Family–Preschool Interaction”. I consider the new version ready to be published and I conrats the authors for the work done.